# Biology of NSCLC: Interplay between Cancer Cells, Radiation and Tumor Immune Microenvironment

**DOI:** 10.3390/cancers13040775

**Published:** 2021-02-12

**Authors:** Slavisa Tubin, Mohammad K. Khan, Seema Gupta, Branislav Jeremic

**Affiliations:** 1MedAustron Ion Therapy Center, Marie Curie-Straße 5, 2700 Wiener Neustadt, Austria; 2Department of Radiation Oncology, Winship Cancer Institute, Emory University School of Medicine, 1365-C Clifton Road, Atlanta, GA 30322, USA; m.k.khan@emory.edu; 3Lombardi Comprehensive Cancer Center, Georgetown University Medical Center, Washington, DC 20057, USA; sg1335@georgetown.edu; 4Research Institute of Clinical Medicine, 13 Tevdore Mgdveli, Tbilisi 0112, Georgia; Info@toduaclinic.ge

**Keywords:** non-small cell lung cancer, immunosuppression, immunostimulation, immunotherapy, radiotherapy, abscopal effect, bystander effect

## Abstract

**Simple Summary:**

The immune system represents an important link for tumor development, tumor control and tumor progression. The tumor immunogenic balance, determined by the prevalently immuno-inhibitory tumor- and conventional radiation-related effects is shifted negatively towards immunosuppression, which can worsen treatment outcome and prognosis. Emerging evidence suggest that those suppressive effects might be converted to an immunostimulative environment that can improve the therapeutic ratio with uses of newer conventional radiotherapy approaches combined with emerging immunotherapy agents.

**Abstract:**

The overall prognosis and survival of non-small cell lung cancer (NSCLC) patients remain poor. The immune system plays an integral role in driving tumor control, tumor progression, and overall survival of NSCLC patients. While the tumor cells possess many ways to escape the immune system, conventional radiotherapy (RT) approaches, which are directly cytotoxic to tumors, can further add additional immune suppression to the tumor microenvironment by destroying many of the lymphocytes that circulate within the irradiated tumor environment. Thus, the current immunogenic balance, determined by the tumor- and radiation-inhibitory effects is significantly shifted towards immunosuppression, leading to poor clinical outcomes. However, newer emerging evidence suggests that tumor immunosuppression is an “elastic process” that can be manipulated and converted back into an immunostimulant environment that can actually improve patient outcome. In this review we will discuss the natural immunosuppressive effects of NSCLC cells and conventional RT approaches, and then shift the focus on immunomodulation through novel, emerging immuno- and RT approaches that promise to generate immunostimulatory effects to enhance tumor control and patient outcome. We further describe some of the mechanisms by which these newer approaches are thought to be working and set the stage for future trials and additional preclinical work.

## 1. Introduction

The World Health Organization (WHO) estimated 1.76 million deaths caused by non-small cell lung cancer (NSCLC) in 2018. Lung cancer represents the leading cause of deaths worldwide with NSCLC representing approximately 85% of all lung malignancies [1]. Despite substantial improvements in therapy, 5-year overall survival (OS) for NSCLC does not exceed 25% [2,3]. Prognosis and survival of patients affected by NSCLC are correlated to disease stage, with OS being prognostically favorable by earlier diagnosis and treatment [4,5]. Nevertheless, a significant percent of patients are diagnosed in advanced stage with little chance to be cured.

Chronic inflammation plays a key role in tumorigenesis of NSCLC [6]. Inflammatory factors like cigarette smoking are associated with chronic bronchitis and emphysema, which lead to the development of lung cancer [7]. The process of cellular malignant transformation consists of many steps over long time periods, ranging from pre-carcinogenic chronic inflammation, progressing, if not treated, towards the development of invasive carcinoma and systemic disease [8,9,10]. Some chronically inflamed pre-carcinogenic environments will never result in malignant cell transformation, while others exposed to the same carcinogen will undergo tumorigenesis. This uncertainty of initiating tumorigenesis events highlights the malignant “modulating” role of genetic predisposition in cancer [11] and underscores that inflammation may assist in this process. The profile and status of the inflammatory-changed environment following the chronic carcinogen(s) exposure is exceedingly variable, and its potential for malignant transformation is related to polymorphic immune response genes affected by a diversity of anti-oxidant and DNA repair associated genes [11]. The immune system, in its cell-mediated and humoral form, is deeply involved in generation of an inflammatory environment and is considered to be the first step of tumorigenesis. Chronic exposure of normal cells to carcinogen(s) leads to initiation of immune cell activation with subsequent upregulation of the pro-inflammatory cytokines like Interleukin-1 alpha (IL-1α) and IL-1-β, and production of Cyclooxygenase (COX)-1 and COX-2 immune-regulatory enzymes in epithelial and mesenchymal. This activation of the inflammatory pathways is associated with the development of malignant disease [12,13,14,15]. Additionally, the induction of COX-2 favors increased angiogenesis in an inflammatory environment and helps pave the way for hyper-vascularization needed for tumor development and progression [16,17,18].

In addition to assisting tumorigenesis, the immune system also plays a fundamental role in the defense against cancer by surveillance and identification of foreign or “non-self” from self and assisting with elimination of DNA-damaged cancer cells from the body. However, tumor cells can escape by suppressing the immune system. Furthermore, therapies, especially if they cause immune-suppression, may further aid the escape of the tumor cells from the immune system. Most patients with NSCLC will eventually require radiotherapy (RT) alone (especially those that are medically inoperable) or in combination with systemic therapy (especially those with advanced stage disease). Regardless, there are many reports that show that conventional RT induces an immunosuppression, and thereby can negatively affect the overall survival [19,20,21,22]. Thus, the current immunogenic balance, determined by the inhibitory effects caused by tumor cells themselves and radiation therapy can lead to poor clinical outcomes. There is newer emerging evidence suggesting that immunosuppression is an “elastic process” that can be transformed into an immune-stimulating environment by “correcting” many manipulable components of this triangle radiation, tumor, and immune cells by using combination approaches that change the way RT, chemotherapy, and immunotherapy are used. Significant efforts are being deployed in the development of novel treatment strategies and protocols aimed to convert radiation- and tumor-induced immunosuppression into a predominant immunostimulation by means of immunotherapy, unconventional RT and the combination of these modalities in order to improve the outcomes of NSCLC patients. The aim of the present review is, therefore, to highlight the natural interplay between the NSCLC cells, tumor immune microenvironment and radiation that routinely results in immunosuppression, and how those immunosuppression-related factors could be manipulated for generation of prevalently immunostimulatory effects to improve NSCLC patient’s outcome.

The following two sections will focus on evidence indicating the tumor- and radiation-related immunosuppressive effects, followed by discussions about novel strategies for overcoming and converting these effects into immunostimulative effects.

## 2. Tumor-Related Immune Suppression in NSCLC

The interplay between the tumor and immune cells is the subject of an extended and ongoing research. Table 1 represents latest evidence on tumor-related immunosuppressive effects [23,24,25,26,27,28,29,30,31,32,33,34,35,36,37,38,39,40,41,42,43,44,45,46,47,48,49,50,51,52,53,54,55,56]. Tumor cells escape immune surveillance by downregulation of HLA and co-stimulatory molecules, or by production of immunosuppressive factors and upregulation of immune cell apoptosis inducing molecules [23,24,25,26]. As a consequence, the immune system will “ignore and tolerate” tumor cell proliferation and progression. The presence of tumor-infiltrating lymphocytes (TIL) in cancer cell nests is an independent prognostic factor of survival in various types of cancers including NSCLC [57], and if used properly, could be used to help convert immune cells to fight against cancer. The inefficiency of the immune response against tumor is inversely proportional to tumor growth, being weaker in larger tumors and stronger in smaller tumors [58]. There is a gap in the current knowledge and understanding of the mechanisms behind the immune response against NSCLC, and additional attention to these mechanisms are needed if we are to improve the outcome of patients in the future.

The currently available data comes from peripheral blood or surgically removed NSCLC tumor tissue, with the latter being limited to less than a third of operable NSCLC patients. The vast majority of the patients have an unresectable disease or are inoperable and undergo radio/chemotherapy, and therefore NSCLC tumor tissue in these advanced stage patients is typically unavailable for detailed immunological analysis. However, based on the limited histological and immunological analysis obtained from the available tissue, the main components of a tumor-directed immune response are represented by a complex interaction between various immune cells operating a composite cytokine network that is supported by the surrounding mesenchymal, epithelial and endothelial cells. The immune cells are a large, highly cooperative family consisting of tumor infiltrating lymphocytes (TILs), the tumor associated macrophages (TAMs), the tumor associated neutrophils (TANs), tissue eosinophilia, and T-cell lymphocytes [59,60,61,62,63,64]. The lung anti-tumor immune response is imitated by activation of the pulmonary antigen presenting cells (APCs), represented by macrophages and dendritic cells [65]. This is a fundamental step towards the beginning of an effective anti-tumor immune response. Following tumor antigen(s) recognition and distinguishing “self” from “non-self”, the APCs migrate to the regional lymph nodes and activate the effector immune cells that aid in destruction of tumor cells. These effector immune cells, also known as cytotoxic lymphocytes, include the CD4+ lymphocytes, natural killers, natural killer T-cells, CD8+ lymphocytes and B lymphocytes [66,67,68]. The activation of these cells are enhanced by the secretion of inflammatory cytokines such as IL-12 and Interferon gamma (IFN-γ). These cytokines are released by the activated macrophages, growing tumor cells, and stromal cells surrounding the tumor. Additionally, membrane-receptor induction of programmed death by the cytotoxic lymphocytes also aids cytokine release and apoptosis of tumor cells [69], as the final coordinate anti-tumor response.

For the purpose of effective antigen presentation, a critical role is played by interaction between co-stimulating molecules on antigen-presenting cells and corresponding receptors on cytotoxic lymphocyte [70]. One mechanisms of immune suppression that tumor cells use, are to block this antigen/cytotoxic lymphocyte interaction and prevent the cytotoxic lymphocytes from getting activated against the tumor (Table 1). Several additional tumor-related factors and mechanisms that result in immune suppression have also been described. One of those involved is alterations in signal transduction molecules on effector T-cells leading to the lack of tumor antigen recognition and missing anti-tumor immune response [71]. In this case, increased tumor-related anti-inflammatory humoral factors like IL-10 or Tumor Growth Factor-beta (TGF-β) induce the loss of signal transducer CD3-ε chain (CD3-ε) in TIL. With that, the signaling pathway for T-cell activation is inhibited, the immune response cannot be initiated and results in immunosuppression.

Alteration of CD3-ε, which is involved in tumor-induced T-cell apoptosis, leads to tumor induced caspase-dependent apoptosis in high proportion of tumor infiltrating T-lymphocytes [72,73]. Further, tumor escapes immune control through the process of immune editing, having as the target the loco-regional tumor microenvironment (TME). Several different tumor-related soluble molecules are involved in this form of immunosuppression: Vascular Endothelial Growth Factor (VEGF), Prostaglandin E2, TGF-β, IL-10, soluble phosphatidylserine, MICA Fas and FasL100 [27,28,29,30,31,32,33,34]. Their immunosuppressive effects include inactivation of dendritic cells and T-cells, inhibition of Fas-mediated and NKG2D-mediated killing of immune cells, and release of anti-inflammatory mediators such as IL-10 and TGF-β that inhibit dendritic cells and T-cells [35,36,37,38]. All those effects promote metastatic spread and progression in NSCLC patients [74,75].

Finally, the stromal cells from the TME also exhibit an important immunosuppressive role through modulation and binding of tumor antigens. By binding tumor antigens, these cells compete with the antigen-presenting cells so that many tumor antigens will be down-regulated, resulting in immunosuppression and tumor progression [76,77,78]. By increasing interstitial fluid pressure in the tumor, stromal cells will make significant quantity of tumor antigens to be unavailable and therefore, ignored by T-cells [79]. Besides the tumor and the TME causing immune-suppression, therapies such as convention RT and chemotherapy can also lead to additional immune suppression.

## 3. Radiation-Related Immune Suppression in NSCLC

The wide use of RT in the management of NSCLC, especially for locally-advanced, high-volume disease, is associated with radiation-induced lymphopenia. The radiation-induced lymphopenia has been correlated with poor oncologic outcomes. Table 2 summarizes the many radiation-induced immunosuppressive effects.

Although RT represents a local treatment, it can add to the cytotoxic effects on the circulating immune component as blood flows through the radiation field whose size, together with prolonged treatment times and increased dose fractionation determines the severity of immune depletion [80]. As for tumor cells and healthy cells, radiation is directly detrimental to all cells located within the radiation field. The immune cells are among the most radio-sensitive cells, being easily damaged and killed by ionizing radiation [92,93,94]. Preclinical evidence confirmed that dendritic cells that act as professional APCs and normally responsible for priming of naive T-cells, are significantly damaged after RT. The RT-induced destruction of these dendritic cells can negatively affect T-cell activation [81]. Additional immune cells that are unintentionally targeted, like dendritic cells, macrophages and B- and T-lymphocytes within the irradiated regional lymph nodes, could also contribute to RT-induced immune-suppression. This “collateral” damage of the peri-tumoral immune cells during RT of tumor cells is a result of trying to treat the target volumes that contain tumor (GTV) and surrounding clinical target volumes (CTV) whereby, tumor is likely to have spread. The collateral damage from scattered irradiation can cause damage to APCs, lymphocytes, supportive mesenchymal and epithelial cells as a bystander consequence of trying to irradiate the tumor volumes. Furthermore, conventional RT volumes are expanded to create larger volumes (i.e., internal target volume (ITV)) to account for the tumor motion due to respiration, and taking into consideration tumor motion in 4D-CT, and the planning target volume (PTV) (to account for the set up error(s) related to the patients daily (re)positioning) (Figure 1). The final target volume for radiation treatment will therefore include much larger area than the one corresponding to the macroscopic lung cancer with significant amount of the peri-tumoral tissue between the GTV and PTV, which will be exposed to the full radiation dose. Thus, there is substantial normal/surrounding peri-tumor microenvironment that is irradiated during conventional irradiation approaches (Figure 1), and could result in destruction of many surrounding immune cells circulating within the irradiated volumes.

Furthermore, if the regional lymph nodes are also included in the target volumes, irradiation will further extend the radiation-induced immunosuppression to the whole anatomical region where T-cell priming is expected to take place. The bigger the treatment volume during RT, the greater the inhibitory effects of radiation on the immune system. Further, an extended treatment time in terms of normo-fractionated RT (in order of several weeks, typically 5 in case of (neo) adjuvant and 6–7 in case of radical treatment) will additionally increase the radiation-immunosuppressive power. It has been confirmed that radiation-related lymphocytes depletion following RT for NSCLC associates with poor oncologic outcome, indicating that large radiation volumes and multiple daily fractions leads to systemic immunosuppression [19]. Without a doubt, RT destroys tumor cells and results in cure of low-volume, earlier stage lung tumors, with probably no significant immunosuppressive impact on patients’ prognosis. However, for those patients whose disease is large and more advanced, and requires radiation volumes that are very large, the radiation-induced immunosuppression might have significantly higher impact and relevance for their prognosis. Furthermore, for these patients with more advanced disease, RT is usually given in combination with chemotherapy and the combined effects of these therapies will further exacerbate systemic inhibitory effect on the immune system. The consequences include compromised immune priming, lymphopenia and finally weak anti-tumor immune potential. Indeed, higher radiation doses to the immune system following the definitive RT for stage III NSCLC patients were associated with increased tumor progression and death [82]. One study found that the lower the lymphocyte loss at 6 months after RT (every 100 lymphocytes/mcL), the greater the improvement on PFS and OS. This finding suggested that lymphocyte depletion during RT reduces tumor control and survival in patients with stage III NSCLC [83], while the opposite is also true. Similarly, a secondary analysis of RTOG 0617, including 464 patients affected by stage III NSCLC, found that increased radiation dose to the immune cells was highly prognostic for decreased OS and PFS [84]. These studies suggest that immune cells and organs should be considered as an organ at risk during the radiation treatment planning, and should be spared from radiation, if one is to optimize the peri-tumor immune environment and help convert it from a pro-tumor suppressive environment into an anti-tumor pro-immunogenic environment.

## 4. Therapeutic Strategies to Overcome Tumor-Mediated Immune Suppressive Effects in NSCLC

Similar to the melanoma, head and neck, and mismatch repair (MMR)-deficient colorectal cancers, NSCLC tumors are considered “hot” with significant infiltration of tumors by T-cells and high tumor mutation burden (TMB) [95]. However, inflamed TME is not always associated with favorable prognosis. Recently, it has been reported that while inflamed TME is associated with favorable patient outcome in case of lung adenocarcinoma (LUAD) subtype of NSCLC, it is not true for lung squamous cell carcinoma (LUSC) [96]. This difference was attributed to increased expression of immune checkpoint marker expression in immune-inflamed LUSC compared to inflamed LUAD. Moreover, T-cells become exhausted during the attack on the tumor due to constant exposure to the antigens and through immune checkpoint signaling. Therefore, immunotherapy of NSCLC using immune checkpoint inhibitors (ICIs) was started to block this signaling and has become an important cornerstone of NSCLC therapy. However, as explained above, not all NSCLC tumors respond to ICIs as they develop resistance mechanisms due to the constantly evolving interactions between cancer cells and other cells in TME such as other immune cells, cancer-associated fibroblast, and tumor endothelial cells [97]. Therefore, multiple strategies have been developed over time to treat ICI-refractory NSCLC.

The first approach used is the combination of two ICIs, generally anti-PD-1 and anti-CTLA-4 to enhance anti-tumor immune-mediated response. An improved median and 2-year OS over chemotherapy was observed in NSCLC patients treated with nivolumab plus ipilimumab (CheckMate227) [98]. However, treatment-related serious adverse events of any grade were more frequent in the patients treated with combination ICI than with chemotherapy although grade 3 or 4 treatment-related adverse events were similar. In addition to PD-1 and CTLA-4, other immune checkpoints such as LAG-3, TIM-3 and TIGIT have been tested in trials for combination ICI therapies. For example, in a recent CITYSCAPE phase 2 trial, anti-TIGIT, tiragolumab when combined with anti-PD-L1, atezolizumab resulted in a significant benefit in PFS and ORR in PD-L1-positive metastatic NSCLC patients compared to anti-PD-L1 monotherapy [99]. In addition to the combination ICI therapy, ICI treatment as a re-challenge has also been investigated [100]. However, although the responses have been improved following these strategies, tumor-mediated immune suppressive effects still limit the durability and maximal positive outcomes.

In addition to the tumor-mediated increased expression of checkpoints, VEGF and IDO secreted in the TME serve as important immunosuppressive molecules. VEGF promotes hypoxia-mediated neo-angiogenesis in the TME. VEGF inhibitors can restore normal vasculature to enable immune cell infiltration [101] and provide a rationale for their combination with ICIs to improve outcomes. Indeed, in a recent clinical trial, an improved OS was obtained when atezolizumab was combined with doublet chemotherapy and bevacizumab compared to bevacizumab plus doublet chemotherapy [102]. There are multiple studies that are currently testing this concept [103]. IDO1, IDO2 and TDO2 play important role in tryptophan catabolism, a critical metabolic pathway. IDO1 and TDO2 are overexpressed in several cancer types, including NSCLC and are associated with poor prognosis and resistance to immunotherapy [104]. By depleting tryptophan and increasing kynurenine in the TME, these enzymes enhance immunosuppression as Tregs and MDSCs are generated and proliferation and activation of effector T-cells is inhibited [105]. However, a clinical trial in advanced melanoma patients combining IDO1i epacadostat with pembrolizumab failed [106], most likely due to several flaws in the design such as unselected patient population and insufficient dosing [97]. Therefore, IDOi are still being investigated either in combination with ICIs in NSCLC (NCT02460367) or with other combination partners such as RT and STING agonists. Level of another amino acid, arginine that is essential for lymphocyte proliferation and function, is regulated by arginase 1 and 2 (ARG1/2). Similar to IDO, high expression of ARG1/2 has been found in NSCLC [107] and shown to be associated with poor prognosis. These enzymes are mainly released by MDSCs and macrophages in the TME and hamper T-cell function by lowering production of IFN-γ, TNF-α, and other inflammatory cytokines [108]. Therefore, therapeutic inhibition of ARG1/2 is being employed to enhance anti-tumor immune responses. A phase I/II study in advanced or metastatic solid cancers (NCT02903914), including NSCLC is currently investigating the anti-tumor effects of a small molecule INCB001158 alone or in combination with pembrolizumab.

Several components of adenosine-signaling pathway such as CD73, A2a receptor (A2aR) are overexpressed in variety of cells in the TME. Multiple molecular pathways, including mTOR, MAPK, HIF1-α, and TGF-β regulate expression of CD73 and in turn adenosine [109]. CD73, an ectonucleotidase, generates adenosine (an effective immunosuppressive molecule) by breaking down extracellular ATP [109]. In accordance, high expression of CD73 is associated with poor outcomes in NSCLC [110]. Similarly, high A2aR expression results in an increased binding of adenosine and leads to accumulation of immunosuppressive Tregs, MDSCs, proliferation of cancer-associated fibroblasts, inhibition of effector T-cells, lowering of PD-L1 expression on tumor cells and other anti-immune inhibitory consequences in NSCLC [111]. Therefore, adenosine signaling pathway has been targeted by inhibiting either CD73 or A2aR alone, or in combination with ICIs to overcome tumor-mediated immunosuppressive effects in NSCLC [97].

Recently, neoantigens that are produced due to mutations in the tumor cells have been identified in NSCLC. Because these antigens are unique to the cancer cells and are generally immunogenic, vaccines containing these antigens have been developed [112] and can exploit the benefits of ICIs [113]. Melanoma-associated antigen (MAGE)-A3 is expressed in approximately 32% of NSCLC [114,115]. However, vaccines containing this antigen did not improve PFS or OS [116]. Thus, the combination of these neoantigen-based vaccines with ICIs may be needed and studies in this direction are required. In this regard, it is important to recognize that the sequencing of vaccine and the ICI is important to achieve optimum results. Recently, it is reported that under suboptimally-primed CD8+ T-cell conditions, PD-1 blockade increases the generation of dysfunctional PD-1+CD38hi cells, leading to anti-PD-1 therapy resistance [117]. Accordingly, it may be speculated that treatments such as RT that act as in situ vaccine may be administered before ICI treatment to achieve improved outcomes.

It is now clearly established that TME-mediated immunosuppressive effects hinder the anti-tumor immune response, which is significantly influenced by the heterogeneity of the TME. Targeting these pathways have been employed either alone or in combination with immunotherapies but with limited success, suggesting that other novel, unconventional strategies such as Stereotactic Body Radiotherapy (SBRT)-based PArtial Tumor irradiation targeting HYpoxic clonogenic cells (SBRT-PATHY) may still have scope to further improve treatment outcomes in NSCLC (discussed in the following paragraph).

## 5. Radiation and Immune Stimulation in NSCLC: Bystander and Abscopal Effects

RT, if used appropriately, has a potential to convert a TME into a immuno-stimulative environment that can aid local and distant radiation-induced immune-mediated anti-tumor response [118] and enhance response to immune checkpoint inhibitors [119,120,121]. RT can alter the tumor micro-niche by increasing neo-antigen shedding, increase PD-L1 expression, increase MHC class I expression, and reverse exhausted CD8+ T-cells [120]. RT can also increase infiltration of CD8+ T-cells into the tumor and TME and could potentiate the response [119] and can be thought of as a form of immunotherapy with systemic effects [122]. It can alter tumor cells to increase stimulation or immunogenic cell death pathways within the tumor cells, upregulate MHC molecules within tumor cells, increase release of DAMPs, promote expression of cryptic tumor antigens, and lead to production of immune-stimulatory cytokines and chemokines [85]. RT, if used properly, could also alter APCs to promote infiltration into the tumor cells, improve maturation of APCs, alter the APCs to acquire a more immune-stimulatory phenotype, encourage APCs to uptake antigens and to improve processing cross presentation of antigens by APCs, promotion production of immune-stimulatory cytokines and chemokine production, and aid APCs to enhance migration to regional lymph nodes [85]. At the T-cell level, RT has been shown to increase infiltration into the tumor, especially CD8 and CD4 T-cells, promote production of immune-stimulatory cytokines by the T-cells, help T-cells maintain effector function, and may alter T-regs [85]. Table 3 highlights proposed mechanisms as to why radiation could induce an immuno-stimulatory effect that could enhance response to checkpoint inhibitors.

The above mentioned radiation immunogenic effects are typically observed in the preclinical, experimental conditions but clinically their therapeutic impact remain negligible following the use of conventional RT which is considered to be a weak immune-stimulator. In fact, conventional RT usually shows an immunosuppressive character (Table 2: [19,20,21,22,80,84,85,86,87,88,89,90,91]). Accordingly, the 5-year overall survival rate of NSCLC patients to RT and chemo-RT remains at a dismal low ranging from 68% for stage IB to <10% for stage IVA-IVB NSCLC [123]. These results suggest that a combination with other strategies such as immunotherapy is essential to enhance the response of these radio- and radio-chemotherapy-resistant NSCLCs. The RT would induce activation of the immune system against the tumor cells (as highlighted in Table 3) while its immunosuppressive effects can be reversed by ICIs. Indeed, several clinical trials are currently ongoing to explore the potential of the combined strategy of RT and immunotherapy and have been recently reviewed for both stage III and advanced NSCLC [124]. It is important to note that while in some of these trials, RT and ICIs are being administered concurrently, in others ICIs are given as an adjuvant therapy after RT. The sequence of these therapies are dependent on several factors such as the dose, fractionation, dose/fraction of RT, the type of ICI to be used as well as on the intrinsic properties of the tumor and their response to the first line of therapy as we have reported earlier [121].

In addition to the combination approaches, the immunosuppressive effects of RT could be turned into a pre-dominant immune-stimulatory effect leading ideally to clinically desirable abscopal effect (AE) and bystander effect (BE) using novel, unconventional delivery approaches of RT. BE and AE are tumoricidal non-targeted immune-mediated radiation effects that have a great anti-tumor potential and thus significant clinical relevance. They both represent an out-of-field-extended regression of non-irradiated local (BE) or distant (AE) tumor lesions as a result of an optimally balanced interplay between the radiation-induced, pro-inflammatory and anti-inflammatory cytokines, TME and immune system cells. The characteristics of BE/AE induced by conventional RT are described elsewhere in more detail [125]. To be optimally pro-immuno-stimulative, RT needs to be administered in a way to release enough quantity of hidden tumor (neo) antigens required for a potent immune-stimulation and to maximally spare the loco-regional peri-tumoral immune cells necessary to induce BE/AE. This balance is only occasionally achievable after conventional, normo-fractionated RT, which is a weak inductor of BE and AE. Modern radiation technique needs to be properly adapted and adjusted, if one is to drive the maximum BE and AE effects. 

One of the attempts to make such a novel, unconventional RT is SBRT-PATHY, purposefully developed to improve immunogenic potential of radiation, and to act synergistically with the immune system [118]. This approach is fully subordinated to immunostimulation. The key components of this technique are: (1.) partial tumor irradiation targeting the more immunogenic-hypoxic clonogenic cells, (2.) sparing of the loco-regional immune cells as an organ at risk, and (3.) time-synchronization of irradiation with the homeostatic oscillation of the anti-tumor immune response, giving radiation at certain, individually determined optimal timing corresponding to the most reactive phase of the immune anti-tumor response. Typically, the treatment is given in very short time (1–3 fractions) so as not to interfere much with the functioning of the immune system, using an immunogenic-high radiation dose, in order to be fully stimulative. Although this is an emerging technique and thus only small number of patients have been treated so far, its preliminary results in terms of BE- and AE-induction are encouraging [118]. Most of the patients treated with this approach were affected by unresectable bulky NSCLC, whose immuno-suppressive biological behavior showed to be manipulable resulting in immunostimulation and consequent improvement of treatment outcomes [126]. Indeed, the immunohistochemistry and gene-expression analysis of surgically removed partially irradiated squamous cell and adenocarcinoma NSCLCs following SBRT-PATHY, and non-irradiated but regressing abscopal tumor sites, showed activation of immune system in the radiation-spared TME with very dense infiltration of T-lymphocytes, with more or less pronounced predominance of CD8+ cytotoxic lymphocytes [118]. Furthermore, Apoptosis-inducing Factor (AIF) was highly expressed not only in the partially irradiated NSCLCs, but also in the non-irradiated distant tumor lesions pointing to an induction of tumor apoptosis at all sites (partially irradiated and non-irradiated). Surprisingly and interestingly, the lymphocyte infiltration was absent at non-irradiated distant tumor sites, where AIF was highly upregulated, indicating an alternative radiation-induced activation of apoptosis pathway possibly through cytochrome C [118]. Additionally, analysis of non-irradiated abscopal tumor sites performed by real-time PCR of reverse transcribed mRNAs showed the strongest signals of cell death-regulating signaling molecules IL-6, AIF and TNF-alpha, which had higher expression levels compared to the partially irradiated tumors suggesting an abundance of potentially cell death-inducing signals not only in the partially irradiated NSCLCs but even more so in non-irradiated out of field abscopal sites [118]. These findings indicate that the presence of these signaling molecules at abscopal tumor sites may play an important role in the systemic anti-tumor response modulated by SBRT-PATHY at TME and hypoxic segment of partially irradiated bulky NSCLCs. 

A prospective phase I trial is ongoing, currently recruiting patients, aiming to assess the immunogenic potential and optimal timing of SBRT-PATHY for treatment of unresectable bulky tumors of all histologies and organ-sites [127]. Additionally, another prospective phase I/II trial is currently assessing the potential physical and biological advantages of carbon-ions in form of CARBON-PATHY delivered synchronously with an estimated most reactive phase of anti-tumor immune response considering the homeostatic immune oscillations [128]. Moreover, for the purpose of target delineation, CARBON-PATHY is for the first time planned using hypoxia-specific [64Cu][Cu(ATSM)] PET/CT.

Interestingly, it has been reported that certain RT byproducts, such as tumor exosomes can play a role in BE [129]. Radiation-induced BE and the role of tumor exosomes are described elsewhere [120]. More recent data demonstrated that RT can induce the production of tumor exosomes that contain DAMPs and key proteins that play a role in radiation-induced abscopal response [130]. Following irradiation, tumor exosomes can activate dendritic cells and NK cells, and can lead to tumor growth delay via an NK cell dependent pathway in a fashion analogous to irradiation itself [130,131]. These findings showed for the first time the link between tumor exosomes-related BE and NK cell-mediated radiation-induced AE adding to the credibility that RT, if used appropriately, can be used to enhance anti-tumor immunity. Recent animal models are now being developed in NSCLC to better understand radiation-induced immune-related abscopal effects and to find ways to optimally integrate RT with emerging systemic immune checkpoint blockade agents such as the anti-CTLA-4 (ipilimumab) and anti-PD-1 (nivolumab) [131,132]. These models have been looking at RT and anti-CLTA-4 and anti-PD-L1 immune checkpoint combinations, and have suggested synergism of the combination approach [133,134]. Golden et al. [135] reported a proof of principle phase 2 study demonstrating that granulocyte-macrophage colony-stimulating factor along with RT (35 Gray in 10 fractions over two weeks) lead to 27% abscopal response in metastatic NSCLC, breast and thymic cancer patients. Keynote-001 trials demonstrated that in a subgroup of 24 or 97 patients with metastatic NSCLC who received thoracic RT followed by pembrolizumab, there were statistically significant notable improvements in PFS (6.3 vs. 2.0 months, *p* = 0.008) and OS (*p* = 0.034) compared to those who did not received RT, with notable increase in pneumonitis in those that previously received thoracic RT (13% vs. 1%, *p* = 0.046) [136]. Results from a randomized phase II SBRT trial of sequential SBRT and pembrolizumab alone vs. pembrolizumab also demonstrated an improved overall response rate (41% vs. 19%) and median PFS (6.4 months vs. 1.8 months) in favor of the combined SBRT and pembrolizumab approach.

## 6. Conclusions

Due to a high incidence and dismal treatment outcome, NSCLC represents one of the major research challenges in the 21st century. While the immune system emerged as an important link in the chain of tumor development, tumor control and tumor progression, the immunogenic balance becomes one of the major focuses of future preclinical and clinical work. In particular, researchers are attempting to shift the prevalently immuno-inhibitory tumor- and radiation-related effects towards a more immuno-stimulative one, with hopes of improving the therapeutic ratio that combines optimal RT in combination with emerging immunotherapy agents. RT, if used appropriately, could aid local and distant radiation-induced immune-mediated anti-tumor response and lead to clinically desirable AE and BE. In one such scenario, being increasingly clinically investigated, RT was shaped in such a way to limit unnecessary irradiation of immune cells at the immediate periphery of visible tumor mass. A novel, unconventional RT (SBRT-PATHY) successfully addressed important aspects deemed necessary for the treatment success: partial tumor irradiation targeting possibly the more immunogenic- hypoxic clonogenic cells, sparing of the loco-regional immune cells as an organ at risk, and time-synchronization of irradiation with the homeostatic oscillation of the anti-tumor immune response. Ongoing research with this novel approach will provide a better understanding between the interplay between the host, the tumor, and the various treatment manipulations to render a pro-tumor immune-suppressive environment into an anti-tumor immuno-stimulatory one. This may especially be the case, if novel RT approaches are combined with emerging immunotherapy agents for NSCLC patients.

## 7. Patents

Tubin Slavisa. reported an international patent application PCT/EP2019/052164 published as WO 2019/162050. The authors reported no other conflicts of interest.

## Figures and Tables

**Figure 1 cancers-13-00775-f001:**
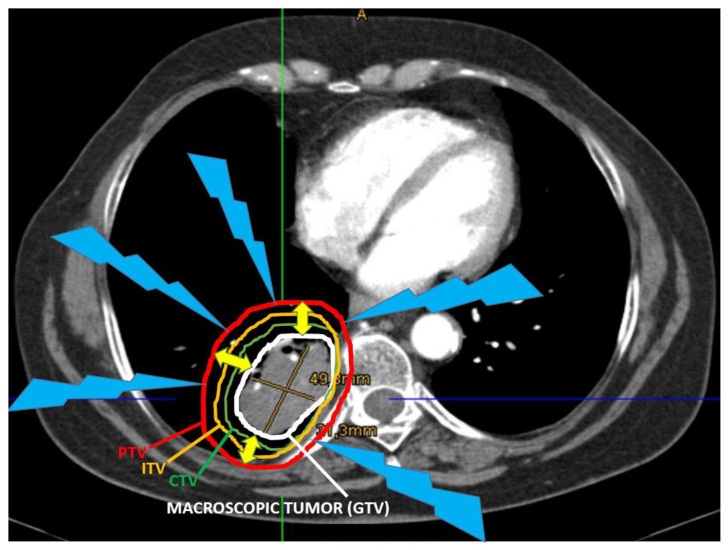
Concept of treating volumes of the conventional RT: the figure shows a centrally located right-side lung tumor (gross tumor volume-GTV, white contour). In order to address the high-risk of subclinical/microscopic disease spread, the clinical target volume (CTV, green contour) is contoured and considered as target for irradiation. Additionally, in order to account for the tumor motion due to respiration, the internal target volume (ITV, orange contour) will also be drawn taking into consideration tumor motion in 4D-CT. Finally, to account for the set-up error(s) related to the patients daily (re)positioning, an additionally larger volume known as the planning target volume (PTV, red contour) is also drawn as the final treating volume for radiation treatment. The yellow arrows indicate the definitive diameter of the final treatment volume in comparison to the macroscopic tumor (GTV). Significant amount of the surrounding peri-tumoral healthy tissue between GTV and PTV will be exposed to the full radiation dose (blue lines), the same that will be delivered to the tumor.

**Table 1 cancers-13-00775-t001:** Tumor-related immunosuppressive effects.

Tumor-Related Effect	Consequences
Production of immunosuppressive factors and upregulation of immune cell apoptosis inducing molecules [23,24].	Escape from immune surveillance.
Downregulation of HLA and co-stimulatory molecules [25,26].	Escape from immune surveillance.
Tumor-related secretion of soluble molecules VEGF, PGE2, TGF-β, IL-10 [39,40,41].	Immune editing, escape from immune control.
Release of anti-inflammatory mediators such as IL-10 and TGF-β [39,40,41].	Inhibition of dendritic cells and T-cells.
Tumor-associated antigens overexpression in NSCLC [42,43].	Immune system tolerance and less responsiveness to immune checkpoint blockade.
Tumor specific (neo) antigens present on MHC molecules, often downregulated in NSCLC [44,45].	Tumor cells evasion of immune destruction.
Lung cancer cells overexpress the immunosuppressive protein, PD-L1 [46,47].	Inhibitory effects on signaling pathways involved in T-cell activation and cytokine secretion.
Tumor cells mediate a checkpoint/“brake” on T-cell activation and thus anti-tumor immunity, by expressing CTLA-4, a B7 ligand and an inhibitory homolog of CD28 [48,49].	Tumor cells evasion of immune destruction.
PD-1’s arrangement with self-ligand PD-L1, found on lung tumor cells dampens the apoptotic pathway [49,50]	Induction of anergy and T-cell depletion.
Tumor cells expand a local immunosuppressive microenvironment, induce dysfunctional T-cell signaling, and upregulate inhibitory immune checkpoints [51].	Evasion of host immune-mediated surveillance and destruction.
Tumor cells express ligands for PD-1 interacting in that way with surface molecules on CD8+ T-cells; influence the microenvironment via orchestration by cytokines [52].	Apoptosis of CD8+ T-cells; immune tolerance.
Tumor cells do not express many neoantigens, and some of those even if expressed might be low immunogenic eliciting only a mild reaction with low affinity antibodies [53].	Cytotoxic lymphocytes unable to recognize tumor cells, inhibited combined cytotoxic reaction together with T-cells.
Release of soluble amino acids tryptophan and arginine within the tumor microenvironment [54,55].	Inhibition of T-cells and NK function, tumor immune tolerance.
Tumor cells express ectonucleotidases CD73 and CD38 which create adenosine from ATP via ADP-AMP [56].	Induction of immunotolerance in cytotoxic lymphocytes.

Abbreviations: Non-small cell lung cancer (NSCLC), Major Histocompatibility Complex (MHC), Cytotoxic T-Lymphocyte-associated Protein-4 (CTLA-4), Programmed Death-Ligand 1 (PD-L1), Vascular Endothelial Growth Factor (VEGF), Prostaglandin E2 (PGE2), Tumor Growth Factor beta (TGF-β), Interleukin-10 (IL-10), Adenosine triphosphate (ATP), Adenosine diphosphate (ADP), Adenosine monophosphate (AMP).

**Table 2 cancers-13-00775-t002:** Radiation-related immunosuppressive effects.

Radiation-Induced Effect	Consequences
Radiation-induced lymphopenia [19,20,21,22].	Immunosuppression.
Cytotoxic effects on the circulating immune component [80].	Immune depletion.
Direct damage of dendritic cells as professional antigen-presenting cells (responsible for priming of naive T-cells) [81].	Negative impact on T-cell activation leading to immune tolerance.
Radiation-induced lymphocyte depletion following RT for NSCLC [19].	Systemic immunosuppression leading to poor oncologic outcome.
Higher radiation doses to the immune system following RT for stage III NSCLC [82].	Systemic immunosuppression leading to increased tumor progression and death.
Radiation-induced depletion of total lymphocytes [83,84].	Reduces tumor control and survival in patients with stage III NSCLC.
Upregulation of the transcription of HIF-1α [85].	Multiple immunosuppressive effects.
Accumulation of immunosuppressive myeloid cells (N2 neutrophils, M2 macrophages, MDSCs) secondary to the increase of CSF-1, SDF-1, CCL2 induced by radiation [86].	Immunosuppression.
Upregulation of adenosine [87].	Multiple immunosuppressive effects.
Accumulation of regulatory T-cells (related to intrinsic higher radio-resistance and increase of immunosuppressive mediators and cytokines induced by radiation) [87,88].	Immunosuppression.
Killing of tumor-infiltrating immune cells (e.g., lymphocytes, APCs) [89].	Immunosuppression.
Upregulation of PD-L1 on cancer cells [90,91].	Inhibition of CTL-mediated tumor killing.
Induction of TGF-β secretion [90,91].	Multiple immunosuppressive effects.

Abbreviations: Antigen Presenting Cells (APCs), Programmed Death-Ligand 1 (PD-L1), Tumor Growth Factor beta (TGF-β), Cytotoxic T Lymphocyte (CTL), Hypoxia-inducible Factor 1-alpha (HIF-1α), Myeloid-derived Suppressor Cell (MDSC), Colony-stimulating Factor 1 (CSF-1), Stromal Cell Derived Factor-1 (SDF-1), CC-chemokine Ligand 2 (CCL2), Radiotherapy (RT).

**Table 3 cancers-13-00775-t003:** Radiation-related immunostimulative effects.

Radiation-Induced Effect	Consequences
Increase of NKG2D ligands, co-stimulatory molecules (e.g., CD80) and adhesion molecules (e.g., ICAM-1, E-selectin) on tumor cells [137].	Enhance recognition and killing of cancer cells by cytotoxic lymphocytes.
Smac release from mitochondria [138].	Increase tumor cells sensitivity to granzyme-induced apoptosis.
Release of chemokines (e.g., CXCL9, CXCL10, CXCL16), increase of adhesion molecules on the vascular endothelium (e.g., VCAM-1), normalization of the tumor vasculature [139].	Facilitate the recruitment of effector T-cells to the tumor site.
Release of ATP * [140].	Release of pro-inflammatory cytokines from APCs (priming of IFN-γ-producing cytotoxic CD8+ T-cells).
Calreticulin translocation to the surface of tumor cells (“eat me” signal) * [141].	Increased tumor cells phagocytosis; Promotes pro-inflammatory cytokines release from APCs.
Generation of novel peptides and increase of the pool of intracellular peptides presented [120].	Increase the anti-tumor immune response.
HSP increase (membrane-bound expression and extracellular release) * [142].	Stimulate innate and adaptive immune responses.
Upregulation of “death receptors” (e.g., FAS/CD95) [143].	Enhance recognition and killing of cancer cells by cytotoxic lymphocytes.
Release of HMGB1 protein (“danger signal”) * [144].	DC migration and maturation (increase in efficiency of antigen processing and presentation) Release of pro-inflammatory cytokines and chemokines from APCs.
Decrease of CD47 surface expression (“do not-eat-me” signal) [145].	Increase tumor cells phagocytosis.
Increased MHC-I expression (critical for antigen recognition by CD8+ TCRs) [146].	Enhance recognition and killing of cancer cells by cytotoxic T-cells.
Accumulation of cytosolic DNA in irradiated tumor cells * [147].	Activation of the cGAS/STING pathway and production of type I IFNs and other pro-inflammatory cytokines (APCs maturation, cross-presentation and T-cell recruitment).

Abbreviations: Natural Killer Group 2D (NKG2D), Intercellular Adhesion Molecule 1 (ICAM-1), CXC-Ligand 9 (CXCL9), CXC-Ligand 10 (CXCL10), CXC-Ligand 16 (CXCL16), Vascular Cell Adhesion Molecule 1 (VCAM-1), Adenosine Triphosphate (ATP), Antigen-presenting Cells (APCs), Interferon (IFN), Heat-Shock Proteins (HSP), High Mobility Group Box 1 (HMGB1), Dendritic Cells (DC), T-cell receptor (TCR), GMP-AMP synthase (cGAS), STimulator of INterferon Genes (STING). * Cellular phenomena related to the “immunogenic cell death” of the tumor cell.

## Data Availability

Data sharing is not applicable to this review article.

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
