# Peer review of "Biology of NSCLC: Interplay between Cancer Cells, Radiation and Tumor Immune Microenvironment"

_cancers, 2021, doi:10.3390/cancers13040775_

Round 1

Reviewer 1 Report

The authors review literature on the interplay between the lung cancer, radiation, and tumor microenvironment. This is an important field of research with many new papers published yearly.

1) Also more than a dozen references are from 2020-2021, the majority of references are from 1978 to 2016, which makes a space to update literature with the focus on 2017-2021. 

2) The tables are very useful for this kind of review. In table I, "trp and arg" could be replaced with the full names, rather than explained in the legend. 

3) Page 4 includes a very long paragraph. Shorter paragraphs facilitate reading and could be beneficial for the presentation. 

4) Figure 1 must be improved or removed. Labels could be made more clear to make the Figure self-explanatory. Overall Figure needs more work in order to be published.

5) Page 9. A long paragraph could be simplified to facilitate the reading.

6) Line 384. Pro-immuno-simulative. Is it the correct word? Was it "stimulative"?

7) Writing and style of page 10 can be improved. In particular, line 436 "nearest neighbor effects"; and lines 440-441, "post-irradiated tumor exosomes" are written twice. 

Author Response

Reviewer 1:

Thank you very much for all constructive comments and suggestions! We took them all in consideration, point by point, in order to improve the quality of our paper:

"1) Also more than a dozen references are from 2020-2021, the majority of references are from 1978 to 2016, which makes a space to update literature with the focus on 2017-2021."

Indeed, we have updated literature by substituting and adding around 30 references published prior 2017, with newer high-quality papers published within the last 4 years. You can see and find that in the references-list.

2) The tables are very useful for this kind of review. In table I, "trp and arg" could be replaced with the full names, rather than explained in the legend.

This has been done!

3) Page 4 includes a very long paragraph. Shorter paragraphs facilitate reading and could be beneficial for the presentation.

We've made few shorter paragraphs from this long one, like you proposed.

4) Figure 1 must be improved or removed. Labels could be made more clear to make the Figure self-explanatory. Overall Figure needs more work in order to be published.

Actually, in my opinion figure 1 was very simple and self-explanatory, showing exactly what we want to be pointed out, which is that the conventional radiotherapy exposes a significant portion of peri-tumoral, surrounding healthy tissues to the full (very high) radiation dose. That figure was schematic representation of the final target that needs to be irradiated-planning target volume (PTV) containing the macroscopic tumor (gross tumor volume-GTV) with surrounding normal tissue. However, we have made another version of figure 1 hoping it will meet your expectations.

5) Page 9. A long paragraph could be simplified to facilitate the reading.

We have improved this paragraph as well by splitting the long one into several shorter.

6) Line 384. Pro-immuno-simulative. Is it the correct word? Was it "stimulative"?

This phrase has been improved.

7) Writing and style of page 10 can be improved. In particular, line 436 "nearest neighbor effects"; and lines 440-441, "post-irradiated tumor exosomes" are written twice.

We have improved that part of the text as well.

Reviewer 2 Report

Current review article entitled,"Biology of NSCLC: interplay between cancer cells, radiation and tumor immune microenvironment" has articulated the effect of radiotherapy on immune cell in tumor microenvironment especially in NSCLC. It is evident from number of studies , which have shown that the irradiation negates immune cell function. Importantly, irradiation to cancer tumor enhance the expression of several Immune cell check point and overexpress surface markers such as PD-1/PD-L1 on cancer cells and tumor infiltrating immune cells and eventually causing immune T-cell tolerance. The number of review articles have addressed and acknowledged the development of immunoradiation therapy in several cancer types. I have few suggestions below.

Can author list recent/ongoing clinical trials of immunoradiation therapy?

Authors need to elaborate the radiation resistant tumor, whether showed response to adjuvant immunotherapy and list few of them. Authors may highlight the therapeutic significance of concurrent immunoradiation therapy and adjuvant immunotherapy after irradiation.  

 63 - 68: the authors need to rephrased the sentence "Chronic exposure of normal cells to carcinogen(s) will lead to initiation of the local immune cell...." 

Line 117, 154, 162, 165, 182, and table 1: TGF-b; Change -b to -β 

Line 356, 357: correct CD8+ T cells 

Line 376: write Table 2.  

Line 415: "Surprisingly and interestingly, the lymphocyte infiltration...." authors need to provide appropriate reference after citation.  

Line 418-423: provide appropriate reference for citation

Table 1/2/3: Authors need to indicate appropriate reference from the study in each row in the table. 

Table 3: what is * representing for ?

Author Response

Thank you very much for all constructive comments and suggestions! We took them all in consideration, point by point, in order to improve the quality of our paper:

Current review article entitled,"Biology of NSCLC: interplay between cancer cells, radiation and tumor immune microenvironment" has articulated the effect of radiotherapy on immune cell in tumor microenvironment especially in NSCLC. It is evident from number of studies , which have shown that the irradiation negates immune cell function. Importantly, irradiation to cancer tumor enhance the expression of several Immune cell check point and overexpress surface markers such as PD-1/PD-L1 on cancer cells and tumor infiltrating immune cells and eventually causing immune T-cell tolerance. The number of review articles have addressed and acknowledged the development of immunoradiation therapy in several cancer types. I have few suggestions below.

Can author list recent/ongoing clinical trials of immunoradiation therapy?

Of course we can, and we did this as it has been suggested.

Authors need to elaborate the radiation resistant tumor, whether showed response to adjuvant immunotherapy and list few of them. Authors may highlight the therapeutic significance of concurrent immunoradiation therapy and adjuvant immunotherapy after irradiation.

We have addressed this suggestion appropriately. See additional text added to the paragraph 5.

63 - 68: the authors need to rephrased the sentence "Chronic exposure of normal cells to carcinogen(s) will lead to initiation of the local immune cell...."

Yes, we have improved this too.

Line 117, 154, 162, 165, 182, and table 1: TGF-b; Change -b to -β

TGF-b has been changed to TGF-β.

Line 356, 357: correct CD8+ T cells

Done.

Line 376: write Table 2.

Done.

Line 415: "Surprisingly and interestingly, the lymphocyte infiltration...." authors need to provide appropriate reference after citation.

Done.

Line 418-423: provide appropriate reference for citation

Done.

Table 1/2/3: Authors need to indicate appropriate reference from the study in each row in the table.

Done.

Table 3: what is * representing for ?

We added the missing text.

Finally, we have also improved the English language.

Round 2

Reviewer 1 Report

The authors addressed the questions from the first revision, which might facilitate the reading of the manuscript. Further clarification of Figure 1 would be beneficial. It is not immediately clear what are the green and blue lines on the top panel of the Figure 1. 

In the version I got (v2) the text is squeezed in several places. It can be a technical issue.